# RsRbohD1 Plays a Significant Role in ROS Production during Radish Pithiness Development

**DOI:** 10.3390/plants13101386

**Published:** 2024-05-16

**Authors:** Qiong Gong, Chaonan Wang, Weiqiang Fan, Shuiling Li, Hong Zhang, Zhiyin Huang, Xiaohui Liu, Ziyun Ma, Yong Wang, Bin Zhang

**Affiliations:** 1College of Life Sciences, Nankai University, Weijin Road 94, Tianjin 300071, China; 1120210518@mail.nankai.edu.cn (Q.G.); listening20aa@163.com (S.L.); 2Tianjin Academy of Agricultural Sciences, Vegetable Research Institute, Tianjin 300381, China; chaonan229@163.com (C.W.); falcon_888@163.com (Z.H.); 13602153199@163.com (X.L.); 3State Key Laboratory of Vegetable Biobreeding, Tianjin Academy of Agricultural Sciences, Tianjin 300192, China; 15022626606@163.com (W.F.); zhanghonggzyx@163.com (H.Z.); 4Tianjin Kernel Agricultural Science and Technology Co., Ltd., Vegetable Research Institute, Tianjin 300381, China; 5College of Life Sciences, Tianjin Normal University, Tianjin 300387, China; ziyunun@126.com

**Keywords:** radish, pithiness, NADPH oxidase, respiratory burst oxidase homologs (Rbohs)

## Abstract

Pithiness is one of the physiological diseases of radishes, which is accompanied by the accumulation of reactive oxygen species (ROS) during the sponging of parenchyma tissue in the fleshy roots. A respiratory burst oxidase homolog (Rboh, also known as NADPH oxidase) is a key enzyme that catalyzes the production of ROS in plants. To understand the role of *Rboh* genes in radish pithiness, herein, 10 RsRboh gene families were identified in the genome of *Raphanus sativus* using Blastp and Hmmer searching methods and were subjected to basic functional analyses such as phylogenetic tree construction, chromosomal localization, conserved structural domain analysis, and promoter element prediction. The expression profiles of *RsRbohs* in five stages (Pithiness grade = 0, 1, 2, 3, 4, respectively) of radish pithiness were analyzed. The results showed that 10 *RsRbohs* expressed different levels during the development of radish pithiness. Except for *RsRbohB* and *RsRbohE*, the expression of other members increased and reached the peak at the P2 (Pithiness grade = 2) stage, among which *RsRbohD1* showed the highest transcripts. Then, the expression of 40 genes related to *RsRbohD1* and pithiness were analyzed. These results can provide a theoretical basis for improving pithiness tolerance in radishes.

## 1. Introduction

Radishes (*Raphanus sativus* L., 2n = 2x = 18) are an important root vegetable crop with rich nutrition, fast growth, and easy cultivation in taproots and leaves. Even though radishes are highly adaptable and do not have stringent soil type and climate requirements, unsuitable planting and growing conditions can lead to physiological disorders in the fleshy roots, such as pithed, hollow, cracked, forked, and malformed roots. Pithiness is one of the main disorders of radish roots, and it is also known as sponginess. It occurs during the ripening and storage period of radishes [1]. The characteristic of pithed roots is a network of white textured stripes, spots, or dull white tissue, which, when observed through a cross-section, forms a sharp contrast with normal tissue [2]. Pithed roots severely affect radish root quality [3].

The formation of pithiness or aerenchyma involves several highly connected processes, including secondary cell wall biosynthesis, programmed cell death (PCD), and lignin deposition. Radish root pithiness was related to the death of xylem parenchyma cells in the root pith caused by the accumulation of reactive oxygen species (ROS) [4]. ROS participate as second messengers in a variety of responses in plant cells [5] and play an active role in many biological processes in plants [6,7,8,9]. At the same time, its formation needs the participation of respiratory burst oxidase homologs (Rbohs). This enzyme is an oxidoreductase that catalyzes the production of superoxide anion (O_2_^.−^) from extracellular oxygen (O_2_) with cytoplasmic NADPH as an electron donor, accompanied by the production of H_2_O_2_ [10,11]. The N-terminal of the Rboh protein contains two EF chiral structures, and the C-terminal contains six α transmembrane helix domains (TMD-I to TMD-VI), FAD, and NADPH [12]. The production of ROS mediated by Rbohs plays an important role in plant development and biotic and abiotic stress responses [13].

Since the *Oryza RbohA* gene was first reported in plants [14], due to the indispensable roles of Rboh proteins in the wide range of biological processes, many members of Rboh-encoding genes have been identified in different plants [15,16,17,18,19]. However, analyses of Rboh systematic genome-wide identification and detailed characterization are lacking in radishes. In the current study, we identified a total of 10 RsRboh genes and used them for the analyses of their important features and functions on radish pithiness. Gene expression analyses performed in five grades of pithiness developmental stages using qRT-PCR revealed their diverse functioning. The study provides comprehensive knowledge of the RsRboh proteins in radishes, which would assist in the precise functional characterization of each gene in future studies.

## 2. Results

### 2.1. Observation of Pithiness Formation

Since pithiness occurs mainly during the maturity and storage period of radishes, we began to observe the pithiness characteristics of radishes at 9 weeks after planting (WAP) in field conditions. None of the radishes showed sponging of the parenchyma tissues at 9 WAP (Figure 1a), however, they started to develop pithing from 10 WAP onwards. Comparing the cross-section of the roots, white spongy tissue in the parenchyma cells of the pithed radish was observed, which pointed to the center of the pith and radiated all over the xylem (Figure 1b). Longitudinally, a white spongy tissue line running through the whole root was visible (Figure 1d). In the most severe cases, the white tissue crumpled to form cavities (Figure 1c), and the fleshy root water content and elasticity decreased, accompanied by a deteriorated taste.

According to the observation, radish pithiness is a process in which the spongy range of parenchyma gradually increases. According to the degree of pithiness of fleshy roots, we classify it into 0, 1, 2, 3, and 4 as five grades, which are denoted by P0, P1, P2, P3, and P4, respectively. We defined the grade of pithiness in P4 as 100%, and compared other stages with P4; thus, quantitatively the pithiness grades were <5, <20, >50, and 100%, respectively, for P1 to P4. (Table 1, Figure 2a).

### 2.2. Relation of Pithiness Formation with ROS

Evans blue was used to dye the dead cells in the root tissue. With the increase in the radish pithiness level, the Evans blue stained area enlarged accordingly, indicating the occurrence of an increased number of dead cells (Figure 2b). DAB (3,3′-diaminobenzidine) is the chromogenic substrate of Peroxidase. In the presence of hydrogen peroxide, it loses electrons and shows color change and accumulation, forming light brown insoluble products. With the increase in radish pithiness, the light brown color deepens after DAB staining, which showed that hydrogen peroxide production increased (Figure 2c). Nitro-blue tetrazolium (NBT) is the substrate of dehydrogenase and other peroxidases, which is often used to detect the existence of superoxide anions. After NBT staining, a blue area occurred in the root with pithiness and darkened along with the development of pithiness (Figure 2d). Observing the parenchyma cells in the radish fleshy roots area under a confocal microscope, a large number of hollowed cells were found to be stacked together, and some cells had broken edges and lost their complete cell contours (Figure 2e). The results of tissue staining and microscopic observation demonstrate that with the aggravation of radish pithiness, the number of dead cells and the production of H_2_O_2_ and O_2_^.−^ increased.

### 2.3. Identification and Characterization of Radish Rbohs

#### 2.3.1. Rbohs’ Protein Information and Genes’ Chromosomal Localization

A total of 10 Rbohs were identified from radish genomic data using a Hmmer search, and after removing redundant, incomplete, and atypical Rboh structural domain sequences. These sequences were submitted to the NCBI and SMART databases for further confirmation of the presence of four characteristic domains such as the NADPH oxidase (Pfam ID: PF08414), ferric reductase (Pfam ID: PF01794), FAD_binding_8 (Pfam ID: PF08022), and NAD_binding_8 domain (Pfam ID: PF08030). Phylogenetic analysis revealed evolutionary relationships among the Rboh proteins of radishes, *Arabidopsis thaliana*, *Brassica rapa*, and *Oryza sativa*. The tree topology indicated six major groups (group I–VI) (Figure 3a). Proteins with high homology are clustered in closer branches. Radish Rboh genes were named according to their evolutionary relationship with AtRbohs (Figure 3a) as RsRbohA—RsRbohJ. Among them, the RsRboh members are not homologous to AtRbohF and AtRbohI. In addition, there are two members that are each homologous to AtRbohD and AtRbohG.

The physical and chemical properties of RsRbohs were analyzed using the ExPASy online analysis tool. The results showed that the number of amino acids encoded by RsRbohs ranged from 611 to 996, the relative molecular mass of the proteins ranged from 69.97 kDa to 112.23 kDa, and the theoretical isoelectric point ranged from 6.87 to 9.95 kDa. The results of chromosomal localization analysis showed that the 10 RsRboh genes were unevenly distributed on seven chromosomes of radishes, but Chr08 and Chr09 were not distributed (Figure 3b). The distribution of the 10 RsRboh genes on the chromosomes was relatively dispersed, with one to three Rboh genes on each chromosome, including one Rboh gene on Chr01, Chr02, Chr03, Chr05, and Chr07, two Rboh genes on Chr04, and three on Chr06. Subcellular localization showed that RsRbohs were mainly located on the plasma membrane, RsRbohE on the plasma membrane and nucleus, and RsRbohG2 on the nucleus, cytoplasm, and vesicles (Table 2).

#### 2.3.2. Gene Structure and Conserved Motif Analyses

An unrooted phylogenetic tree was generated (Figure 4a) and the distribution of conserved structural domains and the exon–intron structures of RsRboh genes and proteins were analyzed. Ten motifs of the Rboh gene family were predicted through the online site MEME (Figure 4b). It was found that RsRbohs contains 10 motifs with largely similar distribution patterns. Motif 2, motif 3, motif 4, and motif 5 are present in all Rboh genes, suggesting that they are highly conserved in the RsRboh family. In addition, RsRbohG2 contains the least number of motifs, while RsRbohJ lacks motif10 and motif1, which are present in all other members. All RsRboh members contained different numbers of exons, indicating a high degree of differentiation among the 10 Rboh genes. In addition, a certain regularity in the length and number of exons was found, with large differences between different branches, but similar gene structures existing in the same branch.

#### 2.3.3. Cis-Acting Element Analysis of Radish Rboh Gene Promoters

The biological functions of the RsRbohs genes were further resolved by predicting cis-acting elements in the promoter region of the RsRboh family (Figure 5). Based on the nature of the Rboh gene family under study, after removing general transcriptional regulatory elements and elements of unknown function, these cis-acting elements were categorized into three main groups, depending on their functions. The first category mainly includes environmental or stress response elements, including the cis-acting regulatory elements necessary for anaerobically induced drought- and high-salinity stress-responsive genes, and drought- and high-salinity stress-responsive genes; cis-acting elements involved in the low-temperature response; MYB-binding sites involved in the drought response; MYB-binding sites; MYC-binding sites; heat-shock protein-associated elements; cis-acting elements involved in defense and stress responses; and wound response elements, stress, and anaerobically induced elements. The second category is hormone response elements, such as salicylic acid and oxidative responsiveness, abscisic acid response elements, growth hormone response elements, methyl jasmonate response elements, gibberellin response elements, ethylene response elements, etc. The third category is growth and development-related action elements, including many light-responsive elements, as well as cis-acting regulatory elements related to the development of meristematic.

### 2.4. Expression of RsRbohs in Different Pithiness Degrees of Fleshy Roots

Tissues from the roots with different levels of pithiness (Figure 6a) were collected for RNA isolation. According to the conserved sequence of RsRbohs, qRT-PCR primers were designed, and the expression levels of 10 RsRboh members in the fleshy roots of radishes with five pithiness grades were obtained (Figure 6b). As can be seen from the results, except for *RsRbohB*, *RsRbohE*, and *RsRbohG1,* the expression of all the other members increased during the early development of pithiness and reached the peak at the P2 stage. Multiple comparisons showed that all RsRboh genes, except *RsRbohC*, were significantly different from each other in the P2 radish. The expression levels of *RsRbohA* at P3 and *RsRbohJ* at P1 were not significantly different from the control, while there were significant differences in the expression levels of other Rbohs in the P0 stage. In addition, changes in the expression level of *RsRbohD1* were the most significant and the level was more than 10-fold higher than that of radishes without pithiness, suggesting that *RsRbohD1* may play the main role in radish pithiness initiation and development.

### 2.5. Comparison of the Expression of RbohD1-Related Genes between P0 and P2 Stages of Radish Roots

To further understand the potential role of *RsRbohD1* in the formation of radish pithiness, based on the previous studies on the function and regulation of *RbohD1* and the production and clearance of ROS, we determined the expression level of 40 related genes. Ten genes involved in the RsRbohD1 were identified based on analyses utilizing SRTING tools (https://string-db.org/, accessed on 7 October 2023) (Appendix A). Fifteen genes are associated with ROS scavenging, and 7 and 8 genes are involved in the phytohormones ABA and ethylene, respectively (Appendix A). As shown in Figure 7, there was a significant increase in the expression of *CPK5* [20,21,22] and *CPK28* [23], *GST* [24], *GPX6* [25], *APX1* [26], *CAT1* [27], *CAT2* [28], *CAT3* [29], *PYL9* [30], *PP2CA* [31,32], *ACS6* [33], and *ERF1A* [34]. In RbohD1-related, Ros scavenging, and ABA-related genes, except for a few genes, such as *CPK4* [35], *GPK6* [36], *APX6* [37], etc., most of them showed an increasing trend. However, half of the ethylene-related genes were slightly higher in the P0 radish than in the P2 level.

### 2.6. Subcellular Localization of RsRbohD1

The *RbohD1-eGFP* fusion expression vector was constructed to observe the subcellular localization of RbohD1. The *OsMCA1-RFP* vector, as the marker of plasma membrane localization, was mixed with the *35S*∷*RsRbohD1-eGFP* vector and *35S*∷*eGFP* empty vector, respectively, and then injected into *Nicotiana tabacum* leaves by the Agrobacterium-mediated method for instantaneous expression. After 2 days of dark culture, the fluorescence signal was observed under the confocal microscope. The fluorescence signal of the empty vector is distributed in the cell membrane, cytoplasm, and nucleus, while the position where *35S*∷*RsRbohD1-eGFP* emits green fluorescence in *Nicotiana tabacum* cells coincides with the position where the plasma membrane localization marker emits red fluorescence, indicating that RsRbohD1 is located in the plasma membrane (Figure 8).

## 3. Discussion

The basic function of NADPH oxidase is to catalyze O_2_^.−^ and NADPH to produce ROS, and participate in inducing host defense genes and protein kinase phosphorylation, activating transcription factors, and activating the ion transport system [38]. In this study, 10 radish Rboh genes have been found. The radish Rboh gene is highly conserved. Conservative motif analysis showed that radish Rbohs shared motif 3, motif 4, motif 2, and motif 5, which represented typical Rboh domains (Figure 4). The RsRboh gene family and their relationship with pithiness were explored. Although the early research discussed the occurrence process and influencing factors of radish pithiness, the study content mostly focused on the changes of secondary metabolites during the pithiness process or the prevention and control methods of field cultivation [39,40,41,42,43], and the research content on the molecular mechanism of radish pithiness was less. Radish pithiness is related to oxidative stress, which leads to the death of xylem parenchyma cells in root pulp [4], which inspires our research. The results of tissue staining and microscopic observation show that radishes do have obvious cell death and ROS production in the process of pithiness (Figure 2). The members of the Rboh family play a key role in ROS synthesis [13], so we studied the members of the Rboh gene family and their relationship with radish pithiness.

Radish pithiness is spongy with parenchyma, which is similar to lysigenous aerenchyma in many dicotyledonous herbs. It involves several highly related processes, including secondary cell wall biosynthesis, PCD, and lignin deposition. In *Arabidopsis thaliana*, the parenchyma cells in the central part of the secondary xylem disappeared and formed air gaps in the hypocotyl 12 weeks after 7 days of flooding when the oxygen concentration was reduced to 4% [44]. In the air space formed in the central part, the cell wall disappeared obviously, indicating that the formation of ventilated tissue is due to cell lysis. In rice, the co-expression of *OsRbohH* with *CDPK5VK* or *CDPK13VK* induces ROS formation, which leads to the formation of lysogenic aeration tissue in rice roots [45]. The expression of *RbohH* is positively self-regulated through ROS signaling mediated by Rboh. This will form a positive feedback loop, leading to the persistent production of ROS mediated by OsRbohH. In the phylogenetic tree, OsRhohH and RsRbohH are on the adjacent branches, suggesting that RsRbohH may have similar functions. Fujimoto used the F_2_ population of the sorghum stem variety and succulent stem variety and localized clone to identify the gene *D* related to pith parenchyma cell death on chromosome 6 [46]. The *D* gene induced PCD in *Arabidopsis thaliana* by activating autolytic enzymes. The fleshy root of radishes changed from a succulent state to a spongy state and eventually produced cavities. During this change, parenchyma cells died, ROS accumulated, and ethylene and calcium signals participated, which was similar to the formation of lytic inflatable tissue.

Radish pithiness is related to oxidative stress, in which related genes participate in this process through ROS detoxification and antioxidation [4]. In their research, the expression of PCD and antioxidant-related genes in two varieties with different pithiness of 9WAP were investigated, and only two *RsRbohD* genes were identified as DEGs in the samples [4]. They think that RsRboh is either not responsible for the production of ROS, or it was induced very early in this case. Because the previous research only involved two kinds of radishes, it did not show the specific trend of Rboh in the process of pithiness change. In our study, the expression changes of RsRbohs in five pithiness levels were described in detail, and it was finally found that the expression trend of most RsRbohs was firstly increased and then decreased, reaching the highest at the P2 level. *RsRbohD1* showed the highest transcripts, which is close to Hoang’s research results. At the P3–P4 level, the number of dead cells increased and the abundance of gene expression decreased.

RbohD regulates the production of cytoplasmic calcium and ROS in plant defense responses, such as ABA signaling to help plants cope with salt stress [47]; the co-regulation of seed germination with ethylene [48]; participation in the accumulation of hydrogen peroxide during cell senescence and death [49]; the production of ROS under hypoxic conditions, which regulates potassium transport with calcium signals [50]; and stomatal closure during drought [51]. Subsequently, superoxide radicals are converted into H_2_O_2_ by the activity of cell wall-localized antioxidant enzymes termed superoxide dismutase (SOD). Among various ROS, H_2_O_2_ is regarded as a signaling molecule in plant stress responses and programmed cell death due to its relatively long half-life, comparably low reactivity, and ability to cross membranes [52,53]. The uncontrolled production of intracellular H_2_O_2_ is sufficient to induce cell death and alter gene expression [54]. Our tissue staining results showed that during the saponification of radish fleshy root parenchyma tissue, the number of dead cells increased, hydrogen peroxide accumulated, and the superoxide anion increased, indicating that ROS was produced in the process of pithiness.

Numerous studies have shown that NADPH oxidase activity in plants may be regulated by various factors including Ca^2+^, receptor-like protein kinases, calcium-dependent protein kinases, and hormones [55]. RbohD may shape the hypoxia-specific Ca^2+^ signatures via the modulation of the apoplastic H_2_O_2_ production [56]. Meanwhile, Ca^2+^ is found to bind RbohD EF-hands and promote ROS production which then subsequently activates calcium channels in response to a range of abiotic stresses [16,57,58,59,60]. Our results show that there are 6 calmodulin kinases among the 10 members associated with the RsRbohD1 protein, among which *CPK5* and *CPK28* are highly expressed (Figure 7). CPKs play a role in signal transduction pathways involving calcium as a second messenger. CPK5 is key for the propagation of the ROS wave in plants [20,21,22], which underlines that Rboh proteins may interfere with Ca^2+^ waves and ROS signaling. CPK28, a regulator of development control, coordinates stem elongation and vascular development [23]. Rboh interaction with calmodulin kinase has also been reported in a number of studies, such as TaRbohs’ interaction with the CIPK26-CBL1 sensor complex, which indicated Ca^2+^-mediated regulation [61]. CPK5 and CPK 6 interacted with and phosphorylated BnaWSR1. The overexpression of phosphomimic *BnaWSR1* in rapeseed protoplasts elicited ROS production and cell death [62]. We speculate that CPK5 and CPK28 may play a role with the RsRbohD1 protein in the process of pithiness, which needs further evidence.

The activity of NADPH oxidase is also affected by hormones. H_2_O_2_ acts as a secondary messenger downstream of the ethylene signal to promote aerenchyma formation under oxygen-deficient conditions [63]. The interaction between ethylene and RbohD was involved in regulating the seed germination and post-germination stage and regulating root growth under normoxic [48]. *Abscisic Acid-Insensitive 4* (*ABI4*), a key component in abscisic acid (ABA) signaling, directly combines with *RbohD* and *Vitamin C Defective 2* (*VTC2*), the key genes involved in ROS production and scavenging, to modulate ROS metabolism during seed germination under salinity stress [47]. Our results showed that genes related to ABA and ethylene change in the process of radish pithiness.

ABA and ethylene play a certain role in the response to abiotic stress, while a similar situation has been reported in *Triticum aestivum*. For example, certain TaRboh proteins are involved in stomatal closure in response to abiotic stress through ABA-mediated pathways, and interactions with other protein phosphatases such as HAB1 and ABI2 are involved in ABA-mediated stress responses [61]. In the non-pithiness and P3 radish, the highest expression level in the abscisic acid signal pathway was *PP2CA*, and *PYL1*, *PYL9,* and *ABF4* were also highly expressed. Key genes related to ethylene signaling, such as *EIN2*, *EIN3*, and ethylene synthase genes, are expressed in varying degrees. A stress environment induces the production or release of ABA, and ABA binds to the receptor PYR1/PYLs/RCARs protein to form a complex, which binds and inhibits PP2C, thus releasing SnRK2 from the state of PP2C inhibition [31,32]. The PYL protein is an ABA receptor. *PYR1/PYL1/2/4/5/8* is highly expressed in stomata, in which *PYL2* dominates stomatal closure induced by ABA, while *PYL4/5* mediates stomatal closure induced by high concentrations of CO_2_ [64]. *PYL9* regulated ABA-induced leaf senescence [30]. In the signal transduction pathway, EIN2 transmits the ethylene signal pathway to EIN3 in the nucleus [65,66]. EIN3 accumulates in the nucleus [67,68] and binds to the promoters of downstream genes to activate the ethylene reaction and promote the production of ethylene. During the ubiquitin degradation of EIN3, EBF1 and EBF2 are responsible for recognizing EIN3 [69]. It is suggested that RsRboh proteins proceed response to the pithiness through an ABA-mediated pathway.

From the histological results, there are many similarities between the pithiness of radish fleshy roots and the formation of the lytic aerenchyma of herbaceous plants; for example, they both have cell death and ROS production, but they are also different. The biological significance of simultaneous tissue formation in *Arabidopsis thaliana* and rice during flooding is to help themselves better adapt to environmental stress and promote tissue metabolism under hypoxia. Compared with the former, the pithiness of radish roots is more similar to the hollow stem of sorghum, because it does not seem to involve hypoxia stress, but occurs when the plant is mature and close to senescence. Our results show that with the increase in the degree of pithiness, cell death, and ROS accumulation, ethylene and abscisic acid participate in it, but the sponge of fleshy root parenchyma is a complex metabolic regulation activity, and we need more evidence to prove the relationship between RsRbohD1and hormone, as well as calcium signal in regulating the pithiness of radish. At the same time, although pithiness is easily affected by the environment, it is also a quantitative trait controlled by heredity. Our study only made a preliminary exploration of the pithiness process of a single radish variety, but the genetic information of this trait is not clear. Afterward, we will create generation-localized populations to reveal key regulatory pathways in conjunction with further delving into the genetics and function of pithiness to guide agricultural production.

## 4. Materials and Methods

### 4.1. Plant MATERIALS Cultivation and Sampling

Among the radish recombinant inbred collection at the Vegetable Research Institute of Tianjin Academy of Agricultural Sciences, we identified a radish line, 20L226, with vigorous growth and the root pithiness phenotype. As a means of understanding and ultimately unraveling radish-releasing pithiness, we sampled fleshy roots of line 20L226 grown under field conditions from 9 to 13 weeks after planting (WAP), which gradually showed an increasing trend of the pithiness grade. The degree of root pithiness is graded on 5 levels (Table 1): P0 (non-pithiness), P1 (partial dryness of tissues), P2 (intermediate, spongy, and small cavities), and P3 (spongy tissues and medium cavities) to P4 (severe pithiness).

### 4.2. Histological Analysis

For cell death staining using Evans Blue, approximately 2 mm thick radish root sections were cut using a slicer and stained in a 1% Evans Blue solution (*w*/*v*) overnight at room temperature (RT; 22–25 °C), after which they were de-stained in 100% ethanol. For H_2_O_2_ detection, root cross-sections were treated with 0.1% (*w*/*v*) DAB (in Tris-HCl, pH 5) for at least 8 h in the dark at RT, washed with ethanol, and then with water. To visualize superoxide anion (O_2_^.−^) accumulation, sections were stained with NBT (0.05% in phosphate-buffered saline buffer, pH 6.1) for 8 h in the dark at RT, washed with ethanol, and then washed with water. The parenchyma was taken from the radish at the occurrence of pithiness, and thin slices about 100 μm thick were cut with a Leica VT1200 Vibratomes (Nussloch, Germany) and observed in Zeiss.LSM710 (Oberkochen, Germany) after a 30 min dark treatment with 0.1 mg/mL of Fluorescent Brightener 28.

### 4.3. Identification of Radish Rboh Family Members

The amino acid sequences of 10 Arabidopsis Rbohs (AtRbohA–AtRbohJ) (http://www.arabidopsis.org, accessed on 7 October 2023) were used as queries to search against the radish genome sequence databases (http://radish-genome.org/, accessed on 7 October 2023) [70]. The radish Rboh sequence was extracted by a blastp comparison in the radish whole genome data using a HMMER search (E-value < 1 × 10^−10^). The output putative Rboh protein sequences were checked by the Pfam database (http://pfam.xfam.org/search#tabview=tab1, accessed on 7 October 2023) and NCBI CD Search tools (https://www.ncbi.nlm.nih.gov/Structure/cdd/wrpsb.cgi, accessed on 7 October 2023) to confirm the presence of the Rboh domains, removing redundant and Rboh-free sequences. The preserved sequences were named based on their homology to members of the Arabidopsis Rboh family. The protein sequences of Rbohs from *Arabidopsis*, *Brassica rapa* (BRAD (brassicadb.cn)), *Oryza sativa* (Phytozome (doe.gov)), and radishes were used for the phylogenetic analysis. The protein sequences were aligned by MAFFT [71]. The unrooted phylogenetic trees were generated using MEGA 11 [72] with the neighbor-joining method and bootstrap with 1000 replicates. Every Rboh protein sequence was uploaded to ExPASy (https://web.expasy.org/protparam/, accessed on 7 October 2023) to compute the amino acid quantity, molecular weight, and theoretical isoelectric point. Wolfpsort (https://wolfpsort.hgc.jp/, accessed on 7 October 2023) was used to predict the location of Rboh proteins on subcellular sites.

### 4.4. Sequence Features and Structural Characterization

The conserved motifs in Rboh proteins were analyzed using the MEME (http://meme-suite.org/index.html, accessed on 7 October 2023) with the following parameters: any number of repetitions and an optimum motif width from 10 to 100 amino acid residues. Motifs with e-values < 1 × 10^−20^ were retained for further analysis. The genomic and coding sequences of radish Rboh genes were submitted to the Gene Structure Display Server (http://gsds.cbi.pku.edu.cn/, accessed on 7 October 2023) to show the exon–intron structures. The results were visualized using TBtools (v2.095) [73].

### 4.5. Chromosomal Locations and Cis-Acting Element Prediction

The chromosomal location of each radish Rboh was retrieved from the Radish Genome Database [56]. Then, the physical map was generated using TBtools (v2.095) [73]. To predict the cis-acting element composition, the 1.5 kb upstream sequence of each radish Rboh from the initiation codon ATG was submitted to Plant CARE (https://bioinformatics.psb.ugent.be/webtools/plantcare/html/) [74]. According to the cis-element types compiled, the prediction results are divided into three categories: “Abiotic and biotic stresses”, “phytohormone responsive”, and “plant growth and development” [75]. After filtering out some elements that are common in all genes, such as TATA-box and CAAT-box, the information was visualized after statistical processing with R-studio (4.3.3).

### 4.6. Total RNA Extraction and Expression Analysis of RsRbohs

The total RNA was extracted by the RNA simple Total RNA Kit (TIANGEN, Beijing, China). Thermo NanoDrop (Waltham, MA, USA) was used to detect the concentration and quality of RNA. RNA was reverse transcribed into cDNA using the TIANGEN (Beijing) Reverse Transcription Kit. All operation procedures followed the manufacturer’s protocols. The acquired cDNA was diluted 5-fold. All the primers were designed using Primer-Premier 5.0 (Premier Biosoft Interpairs, Palo Alto, CA, USA) and synthesized by BGI (Beijing, China). All the experiments set up 3 independent biological repeats, and each biological repeat included 3 technical repeats. We used radish *actin-2* (GenBank No. LOC108862827) as the internal reference gene. The relative expression levels of radish Rbohs were calculated in the 2^−ΔΔCt^ method. The one-way ANOVA of relative expression and multiple comparisons was calculated by SPSS 20 software, and GraphPad Prism 8.0.2 was used to draw the results.

### 4.7. Subcellular Localization

The ORF of *RbohD1* was constructed in the *CaMV35S-eGFP* vector after removing the terminator, and the gene was located upstream of eGFP. The fusion vector was transferred into Agrobacterium tumefaciens GV3101 and the positive monoclones were cultured in a YEP liquid medium containing 50 μg/mL of Kan and 20 μg/mL of Rif at 28 °C. The positive clones were cultured in a YEP liquid medium containing 50 μg/mL of Kan and 20 μg/mL of Rif at 28 °C and then centrifuged to collect the organisms, and the OD600 of the bacterial solution was adjusted to 0.5 with an osmotic solution containing 10 mM of MES, 10 mM of MgCl_2_, 150 μM of AS, and a pH of 5.6. *OsMCA1-RFP* was used as a control for plasma membrane localization [76]. After the injection, the cultures were incubated in the dark for 2 d and photographed using Zeiss.LSM710.

## 5. Conclusions

In conclusion, the present study suggested diverse functions of 10 RsRboh genes in radishes. The gene structure analysis, physicochemical properties of proteins, and phylogenetic analysis revealed their conserved nature. The cis-acting elements and expression profiling suggested the involvement of RsRboh proteins in radish pithiness responses. The current study would support the functional characterization and exploration of the specific role of each RsRboh gene in future studies. These genes might also be useful in genetic engineering for the development of pithiness-resistant radish species in later studies.

## Figures and Tables

**Figure 1 plants-13-01386-f001:**
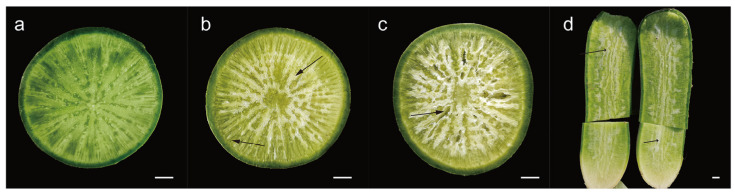
Photos of cross- and longitudinal sections, respectively, of radish roots. (**a**) Without pithiness; (**b**) with moderate pithiness; (**c**) with serious pithiness; (**d**) longitudinal section. The black arrows indicate the presence of white spongy tissue. Scale bar = 1 cm.

**Figure 2 plants-13-01386-f002:**
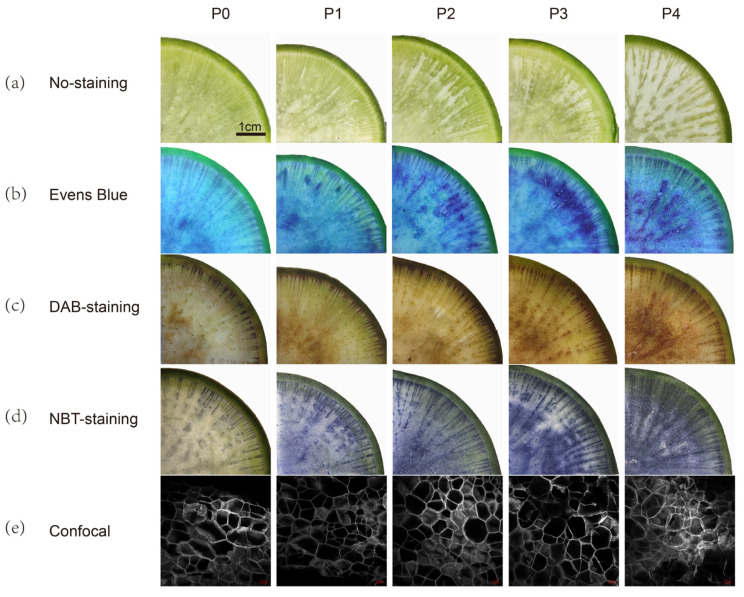
Development of root pithiness in radishes, accompanied by increased ROS production and cell death. (**a**): Roots before staining; (**b**–**d**): roots stained with Evans Blue, DAB, and NBT, respectively; scale bar = 1 cm in (**a**–**d**). (**e**): Confocal scanning, scale bar = 100 μm.

**Figure 3 plants-13-01386-f003:**
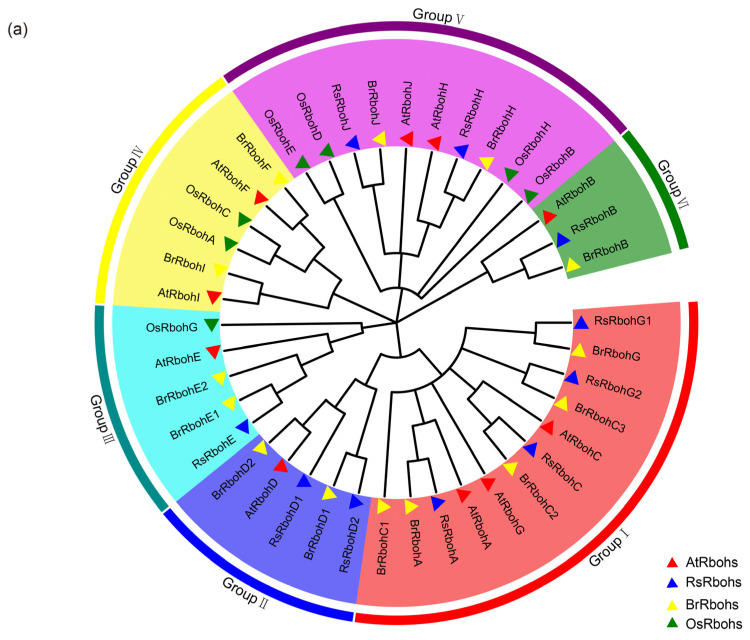
(**a**) Phylogenetic relationships of Rbohs between *R. sativus*, *A. thaliana*, *B. rapa*, and *O. sativa*; (**b**) location of *RsRboh* family members on chromosomes.

**Figure 4 plants-13-01386-f004:**
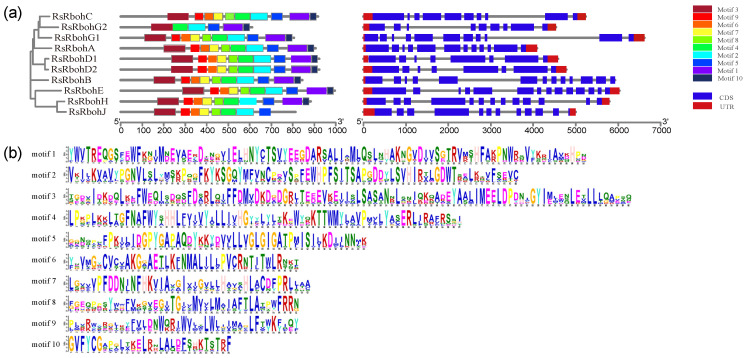
(**a**) Protein motifs and gene structures of RsRboh family members; (**b**) 10 discovered motifs of RsRbohs.

**Figure 5 plants-13-01386-f005:**
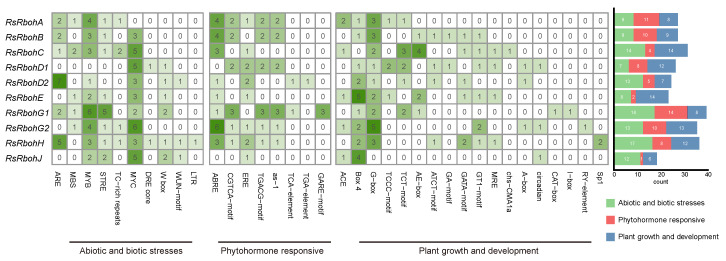
Cis-acting element analysis of radish Rboh gene promoters.

**Figure 6 plants-13-01386-f006:**
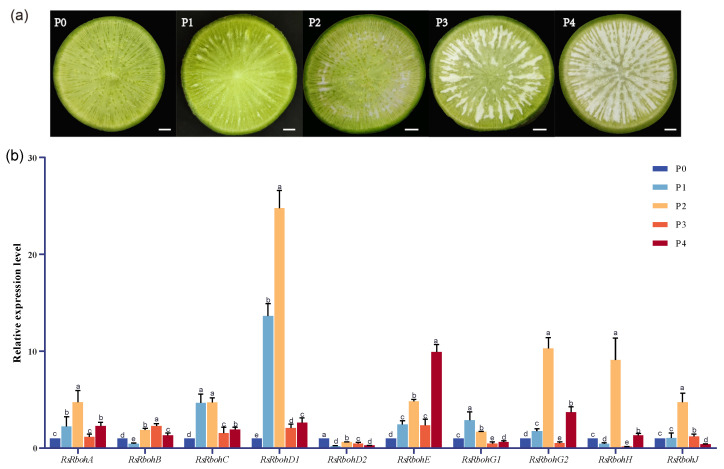
Expression of RsRboh member genes in radish roots with different degrees of pithiness. (**a**): Tissue cross-section, showing the gradual increase in radish pithiness, scale bar = 1 cm; (**b**): relative expression of genes. The expression level in P0 radish was defined as 1.00 (normalized with respect to the actin mRNA levels). Error bars show SD (n = 3). Different letters indicate statistically significant differences (Tukey’s honestly significant difference; α = 0.05).

**Figure 7 plants-13-01386-f007:**
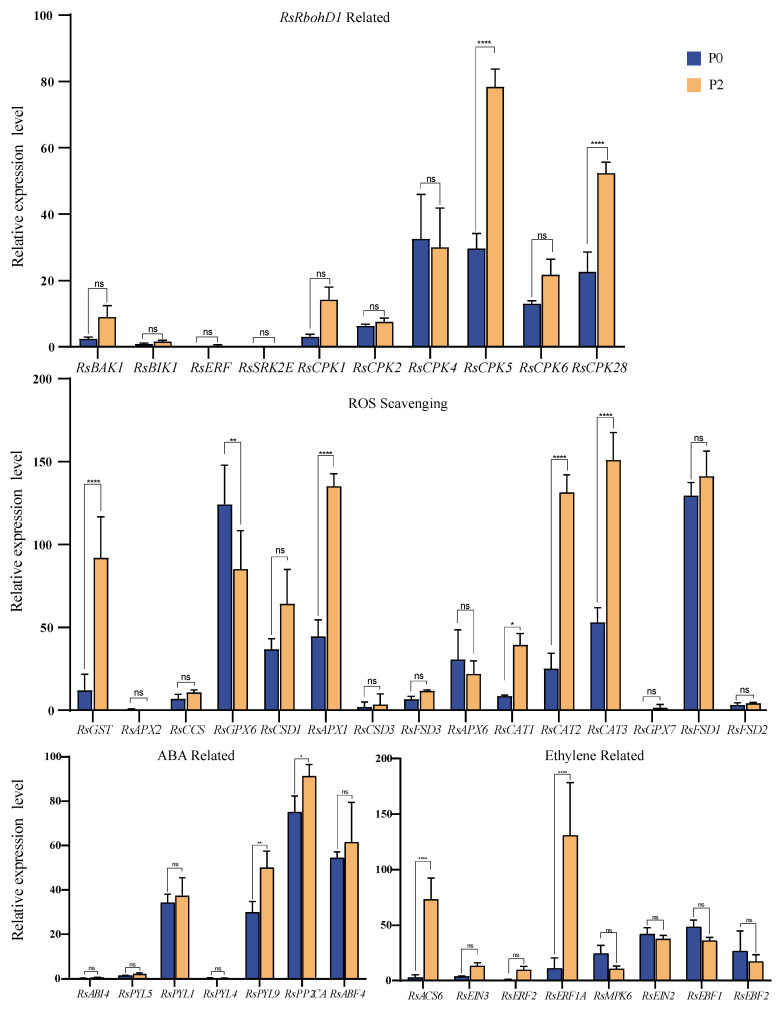
The expression levels of genes related to *RbohD1*, reactive oxygen species scavenging, and plant hormones in radish roots at P0 and P2 levels. Error bars show SD (n = 3). Multiple comparative significant difference; **** *p* ≤ 0.0001, ** *p* ≤ 0.01, * *p* ≤ 0.05 and non-significant (ns) *p* > 0.05, compared with the control (the expression level of *RbohD1* in P0).

**Figure 8 plants-13-01386-f008:**
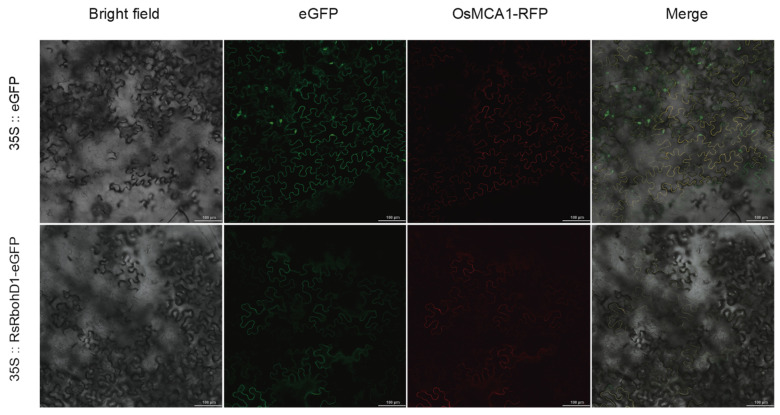
Subcellular localization analysis of RsRbohD1 protein, scale bar = 100 μm. The eGFP and RFP showed green and red fluorescence, respectively, and yellow was the site of co-expression of both.

**Table 1 plants-13-01386-t001:** The 5 grades of radish pithiness and their characteristics.

Pithiness Grade	Parenchyma Tissue Characteristics of Radish Fleshy Roots
P0	no spongy tissue
P1	area of spongy tissue <5%
P2	<20% of tissues are white spongy
P3	>50% of tissues are white spongy
P4	almost completely spongy

**Table 2 plants-13-01386-t002:** Rboh family members in *Raphanus sativus* L.

Gene ID	Gene	Chromosome No.	Protein Length (aa)	MW (kDa)	PI	Subcellular Localization
*Rs392300*	*RsRbohA*	R7	906	103.1579	9.95	Plasma membrane
*Rs256300*	*RsRbohB*	R5	844	96.3529	8.7	Plasma membrane
*Rs132220*	*RsRbohC*	R3	917	103.6033	9.88	Plasma membrane
*Rs172640*	*RsRbohD1*	R4	923	103.7353	9.63	Plasma membrane
*Rs315520*	*RsRbohD2*	R6	923	104.2148	9.56	Plasma membrane
*Rs014600*	*RsRbohE*	R1	996	112.2343	9.28	Plasma membrane and nucleus
*Rs157980*	*RsRbohG1*	R4	805	92.3545	9.76	Plasma membrane
*Rs336320*	*RsRbohG2*	R6	611	69.9732	6.87	Nucleus; cytoplasm and vacuole
*Rs079080*	*RsRbohH*	R2	883	100.2423	9.56	Plasma membrane
*Rs344270*	*RsRbohJ*	R6	812	91.9631	9.41	Plasma membrane

## Data Availability

All data supporting this study are included in the article.

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
