# Peer review of "RsRbohD1 Plays a Significant Role in ROS Production during Radish Pithiness Development"

_plants, 2024, doi:10.3390/plants13101386_

Round 1
Reviewer 1 Report
Comments and Suggestions for Authors
Dear the editor,
The submitted paper describes radish pithiness development and provide data on histological analyses and gene expressions.
Although the experimental results are interesting, some description seems to be insufficient for understanding the provided data.
First, definition of staging in radish pithiness development is most important in this paper in terms of understanding the results. The authors should add more description on the staging, P0-P4. I guess that P0-P4 stages are continuous. It would be good idea to define the quantitative difference between each stage. In figure 6a, it does not easy for me to distinguish differences between the P1 section and P2 section.
Second, it should be explained how the authors obtained P0 – P4 materials. The pithiness development is just related to a growth period? Are any other environmental conditions are associated with?
In figure 3a, explanation on how the phylogenetic tree was constructed and what the value in the tree should be added.
If the authors used gene name such as RsRbohD1 in the manuscript, it is better to use the same gene name in Figure 5.
Related to Figure 6, the authors need to consider multiple comparison in the statistical test.
In Figure 6, definition of relative expression level “ 1 “ must be explained.
In figure 7, I could not catch the meaning of the sentence “The expression level of RbohD1 in P0 served as control.”
Related to Figure7, the references of analyzed genes should be added in result section. In addition, the supplemental table listing each gene displayed in Figure 7 and each reference may be useful for the readers.
Author Response
Dear reviewer,
Thank you for your comments!The point-by-point response to your comments has been upload to attachment.Please see the attachment.

Reviewer 2 Report
Comments and Suggestions for Authors
Review of the manuscript by Gong et al. After reading the manuscript, I doubt that it is suitable for publication in its current form.
The title of the paper should be changed. It is not only the RsRboh1 isoform that significantly increases its expression, but many others also change.
An abstract is not a logical whole. It is a combination of several independent sentences. It lacks a main idea (like the whole manuscript). There are abbreviations in the abstract without knowing what they mean (e.g. P2 stage, it is not explained what it means - you can only read it at the end of the paper in the methodology).
The introduction does not introduce the reader to the topic. There is no purpose to the work. The introduction gives the impression that each paragraph of the manuscript is written by a different author, so the abbreviation PCD is explained twice (in two different paragraphs), as if someone hadn't read and checked the whole work. The explanation of this abbreviation appears unnecessarily later in the paper, e.g. in the discussion. Some names come out of nowhere and what they mean (e.g. RsNAC013).
The name of NADPH oxidase is spelt differently: Rboh and RBOH. This should be standardized in the text. In the discussion, it is written randomly.
I have no comments or objections to the methodology used in the paper and the results obtained. However, in the description of the results, there are abbreviations under the figures that are not explained (e.g. P0-P4).
In general, after reading the manuscript, one gets the impression that these are very preliminary and random analyses. There is no guiding theme for the research carried out. This aspect of the work should be improved before publication and the reader should be interested in the reason for the research.
Author Response
Dear reviewer,
Thank you for your comments!The point-by-point response to your comments has been upload to attachment. Please see the attachment.

Reviewer 3 Report
Comments and Suggestions for Authors
The Ms seems to be good but needs major revision before acceptance.
The introduction needs to be revised and updated with the latest references for RBOHs (such as https://doi.org/10.1016/j.cpb.2023.100315) and other information. The last para should explain the work which has been done in this Ms.
The figures quality should be significantly improved.
Phylogeny should be done with the sequences of a few more species.
The discussion also needs to be updated.
Comments on the Quality of English LanguageMinor editing required
Author Response

(The authors gave the same response as above.)

Round 2
Reviewer 3 Report
Comments and Suggestions for Authors
It may be accepted